# Interstitial lung disease in a veterans affairs regional network; a retrospective cohort study

**Armando Bedoya[1], Roy A. Pleasants[2], Joel C. Boggan[1,3], Danielle Seaman[4], Anne Reihman[1], Lauren Howard[5], Robert Kundich[3], Karen Welty-Wolf[1,3], Robert M. Tighe** [1,3]*

**1** Department of Medicine, Duke University, Durham, North Carolina, United States of America, **2** Division of Pulmonary Diseases and Critical Care Medicine, University of North Carolina at Chapel Hill, Chapel Hill, North Carolina, United States of America, **3** Department of Medicine, Durham VA Medical Center, Durham, North Carolina, United States of America, **4** Department of Radiology, Durham VA Medical Center, Durham, North Carolina, United States of America, **5** Department of Biostatistics & Bioinformatics, Duke University, Durham, North Carolina, United States of America

\* robert.tighe@duke.edu

## Abstract

### Background

The epidemiology of Interstitial Lung Diseases (ILD) in the Veterans Health Administration (VHA) is presently unknown.

### Research question

Describe the incidence/prevalence, clinical characteristics, and outcomes of ILD patients within the Veteran's Administration Mid-Atlantic Health Care Network (VISN6).

### Study design and methods

A multi-center retrospective cohort study was performed of veterans receiving hospital or outpatient ILD care from January 1, 2008 to December 31st, 2015 in six VISN6 facilities. Patients were identified by at least one visit encounter with a 515, 516, or other ILD ICD-9 code. Demographic and clinical characteristics were summarized using median, 25th and 75th percentile for continuous variables and count/percentage for categorical variables. Characteristics and incidence/prevalence rates were summarized, and stratified by ILD ICD-9 code. Kaplan Meier curves were generated to define overall survival.

### Results

3293 subjects met the inclusion criteria. 879 subjects (26%) had no evidence of ILD following manual medical record review. Overall estimated prevalence in verified ILD subjects was 256 per 100,000 people with a mean incidence across the years of 70 per 100,000 person-years (0.07%). The prevalence and mean incidence when focusing on people with an ILD diagnostic code who had a HRCT scan or a bronchoscopic or surgical lung biopsy was 237 per 100,000 people (0.237%) and 63 per 100,000 person-years respectively (0.063%).

**Data Availability Statement:** All relevant data are within the manuscript and its Supporting information files.

**Funding:** This research was supported by an investigator-initiated grant from Boehringer Ingelheim Pharmaceuticals, Inc. (BIPI). BIPI had no role in the design, analysis or interpretation of the results in this study. BIPI was given the opportunity to review the manuscript for medical and scientific accuracy as it relates to BIPI substances, as well as intellectual property considerations.

**Competing interests:** Dr. Tighe discloses that he has received funding from the NIH and industry to perform research studies unrelated to the present study. Dr. Tighe also has served on an advisory board for Boehringer Ingelheim. None of the other authors have any conflicts of interest to report for the submitted work. These disclosures do not alter our adherence to PLOS ONE policies on sharing data and materials.

The median survival was 76.9 months for 515 codes, 103.4 months for 516 codes, and 83.6 months for 516.31.

## Interpretation

This retrospective cohort study defines high ILD incidence/prevalence within the VA. Therefore, ILD is an important VA health concern.

## Introduction

Interstitial lung diseases (ILDs) are a rare group of heterogeneous respiratory disorders characterized by progressive infiltration of the interstitium by immune cells and matrix producing fibroblasts, ultimately leading to development of fibrosis. While idiopathic pulmonary fibrosis (IPF) is associated with the most substantial morbidity and mortality among ILDs, the epidemiology of others that are environmentally associated or secondary to other systemic diseases is less frequently studied. The morbidity and mortality of ILD is dependent on the specific ILD subtype. World Health Organization data of males diagnosed with IPF in the European Union (EU) defined the median mortality at 3.75 per 100,000 people in the EU from 2001–2013 [1]. Additionally, the individual economic impact of ILD is substantial, as analysis of USA Medicare claims showed the total direct cost for patients with IPF was $26,000/person-year between 2001–2008, and the incremental cost over control subjects was $12,124 [2]. Despite the clear impact of ILD on human health [3] and the ongoing efforts to define clinical characteristics, there remain considerable deficits in our understanding of the incidence and prevalence of ILD across various groups.

The Veterans Health Administration (VHA) is one such group where there is a gap in knowledge about epidemiology of ILD. The VHA is the largest integrated health system in the USA and has led epidemiology efforts in other disease processes such as diabetes [4], coronary disease [5], and hypertension [6]. The VA healthcare database, as of 2017, contains more than 9 million subjects [7] and is enriched with older males with smoking histories, which are known risk factors for increased ILD incidence. Based on this, it has been hypothesized that ILD is more prevalent in veterans' populations. However, to our knowledge there have been no large epidemiological studies on ILD in veterans.

The primary objective of this study is to describe the incidence/prevalence, clinical characteristics, and outcomes of patients with ILD who received care within the Veteran's Administration Mid-Atlantic Health Care Network.

## Methods

This was a multi-center retrospective cohort study including patients who received inpatient or outpatient ILD care from January 1, 2008 to December 31st, 2015 at six VHA medical facilities and associated clinics in North Carolina (Asheville, Durham, Fayetteville, Salisbury) and Virginia (Richmond, Salem). Subjects were identified by any single visit encounter coded with either a 515, 516, or other ILD ICD-9 code (135, 501, 508.1, or 518.89) (S1 Table). The selection of these codes was based on a review of the available ILD ICD-9 diagnostic codes. During the study period on 10/01/2015 there was a transition to ICD-10 coding. ICD-10 codes were not used to capture patients during this study period. There were no exclusion criteria. This study was reviewed and approved by the Durham VA Institutional Review Board (IRB #01882/001). The study was performed under a waiver of consent and a waiver of HIPPA as

approved by the IRB. Under the approved protocol, the study team had access to patient identifiers during the performance of the electronic record review. During data extraction performed for the analysis, the data was anonymized and no individual patient identifiers are reported in this study.

Data abstraction used data from the VA Corporate Data Warehouse (CDW) and other electronic medical record (EMR) sources, including VistaWeb. CDW data included demographics, site of visits, ICD-9 diagnostic codes to include pre-selected comorbidities, and dates of procedures (pulmonary function tests, bronchoscopies, radiographs, and surgical procedures). EMR data included smoking history, ILD diagnostic information including thoracic computed tomographic (CT) scans, bronchoscopies, and surgical biopsies, pulmonary function tests, and drug therapies. To supplement CDW data, an ILD specialist, pulmonary specialty pharmacist, and a pulmonary fellow performed EMR abstractions. The ICD-9 diagnosis was confirmed based on a thorough chart review. The data from this chart review included physician notes, and CT scan reports. This was reviewed by the study team to confirm or refute the diagnosis. A random sample of charts (10% of total) were re-reviewed by a separate member of the research team to confirm chart abstraction quality. In instances where patients were coded with both 515 and 516 codes, the study team used the physician notes, exam, laboratory values and CT scan report to clarify which of the ICD-9 codes were accurate and this was then used to define their ICD-9 group. Subject data was collected through January 2019 from the time of first ILD diagnosis until their last visit to the facility, death, lung transplantation, or lost to follow-up. Initial diagnosis date was reviewed and updated if it was earlier than captured in the time range.

The pattern of baseline PFTs using spirometry and DLCO measures were categorized as: 1) restrictive: forced vital capacity (FVC) < 80%; forced expiratory volume in 1 second (FEV1)/FVC > 70%); 2) obstructive: FEV1/FVC ≤ 70%; 3) Isolated DLCO Impartment: FVC > 80%, FEV1/FVC > 70%, diffusing capacity for carbon monoxide (DLCO) ≤ 80%); and 4) normal: FVC > 80%, FEV1/FVC > 70%, DLCO > 80%.

Demographic and clinical characteristics were summarized using median, 25th and 75th percentile for continuous variables and count and percentage for categorical variables. Broad and narrow case definitions were defined. Broad included any individual with an ILD diagnostic code, while narrow required an ILD diagnostic code with evidence of a HRCT scan or a bronchoscopic or surgical lung biopsy. Characteristics were summarized, stratified by ICD-9 code of ILD diagnosis and presence or absence of computed tomography of the chest, surgical biopsy, or transbronchial biopsy and differences were compared using Kruskal-Wallis tests for continuous variables and chi-squared tests for categorical variables. Incidence per 100,000 patients was calculated as the number of subjects diagnosed with an ILD in each year divided by the number of unique patients who visited one of the VA centers in that calendar year. Incidence rates were averaged across years. Date of diagnosis was based on the date of physician note or radiologic study identifying and ILD. Though subjects were initially identified by and ICD-9 billing code from 2008–2015, chart review information was used to define the date of diagnosis and therefore could have been prior to 2008. Prevalence per 100,000 patients was calculated as the number of patients diagnosed with an ILD before December 31st, 2015 and still alive at this date, divided by the number of unique patients who visited one of the VA centers in 2013–2015 and were alive as of December 31st, 2015 (as an estimate of the number of VA patients in the system at these centers). Kaplan-Meier curves were generated based on ICD-9 diagnosis codes. The curves were based on any recorded mortality in the study population through January 2019 when the database collection stopped. A log-rank test was used to test the differences in survival among groups. Statistical analysis was performed using SAS v9.4 (SAS Institute, Cary, NC). A p-value <0.05 was used as the threshold for statistical significance.

## Results

We identified 3293 subjects in our cohort who met our inclusion criteria. S1 Fig illustrates the breakdown of the subjects into "No ILD, 515 ICD 9, 516 ICD 9, or Other ILD" following EMR review. Baseline characteristics for the total population (including the "No ILD" category) are noted in S2 Table. The median age was 69 years-old with a wide age distribution. The majority of individuals are male (96%), white (79%) and current or former smokers (75%). The median BMI was 27.8, with 39% of the population considered overweight and 35% obese. Co-morbid diseases were frequently observed in the cohort. Airway disease was the most frequent reported with chronic obstructive lung disease (COPD; 41%) and asthma (8%) documented in 49% overall. Consistent with this, 29% of the cohort had spirometric obstruction. Interestingly, 65% of total cohort did not have pulmonary function testing recorded in the EMR. Gastro-esophageal reflux disease was also a frequent comorbidity, recorded in 40% of subjects. Lung cancer and mixed connective tissue disease were noted at 8% and 1%, respectively.

Review of the available clinical information in the EMR determined that 879 subjects (26%) had no radiologic or clinical evidence of ILD. Given that this represented a large portion of the cohort, Table 1 segregates characteristics by individual ICD-9 code from those who had no evidence of ILD. The vast majority of subjects had an ICD-9 515 code (47%). Clinical characteristics between the ICD-9 groups were similar, with the exception of COPD, where 49% of 515 coded subjects having this co-morbidity versus only 39% in the 516 grouping. This was reflected in the 515 group spirometry, as they had an increased percentage of individuals with obstruction when compared to the 516 group. Alternatively, the 516 group predominantly exhibited a restrictive spirometric pattern. Subjects with no documented ILD on chart review had similar characteristics to those with ILD except for lower rates of smoking (29% never smokers) and COPD (30%). As the 516 ICD code included several sub-groupings, we stratified by individual 516 ICD subset (S3 Table). Within the 516 group, the majority (81.1%) were coded as 516.31 (Idiopathic Pulmonary Fibrosis), or 516.8 (Other specified alveolar and parietoalveolar pneumonopathies). Interestingly, minimal differences existed in the clinical characteristics or co-morbidities between these 516.31 or 516.8 groupings.

Table 2 reports the differences in clinical care of subjects stratified by ICD code. Overall, the cohort had relatively high rates of high-resolution CT scans performed across both 515 and 516 ICD-9 codes (>79%). Subjects with 515 and 516 codes had similar, but low, rates of bronchoscopy. Across all groups, there were low rates of transbronchial or surgical lung biopsies performed. There was moderate usage of oxygen therapy, which increased from initial diagnosis to the last recorded visit. There was a moderate use of corticosteroids and low rates of IPF therapy use across the cohort.

We then defined the incidence and prevalence of ILD in VISN6. Given our broad entry criteria for chart review, we applied broad and narrow case definitions to this analysis. A broad case definition was defined as any individual with an ILD diagnostic code. The narrow definition was defined as any individual with an ILD diagnostic code who had a HRCT scan or a bronchoscopic or surgical lung biopsy. The overall estimated prevalence in the verified ILD subjects (including 515 and 516) was 256 per 100,000 people (0.256%) with a mean incidence across the years of 70 per 100,000 person-years (0.07%). Prevalence and incidence rates stratified by VISN6 locations and year are in Table 3. Using the individual ICD-9 groupings, 515 had a prevalence of 158 (0.158%) and a mean incidence of 49 (0.049%), while the 516 group were 98 (0.098%) and 22 (0.022%), respectively. To determine the impact of the diagnosis on mortality, we used clinical data from this cohort over the course of their medical care within the VA. Fig 1 illustrates a Kaplan Meier curve comparing subjects with a 515, 516 (excluding 516.31), or 516.31. The median survival was 76.9 months for 515 codes, 103.4 months for 516

**Table 1. Characteristics of patient population stratified by ICD-9 code.**

| | 515 (N = 1552) | 516 (N = 742) | No ILD (N = 879) | Other ILD (N = 120) |
|---|---|---|---|---|
| **Age at diagnosis** | | | | |
| Median (Q1, Q3) | 69 (62, 78) | 69 (62, 79) | N/A | 68 (63, 76) |
| **Gender** | | | | |
| Female | 65 (4%) | 23 (3%) | 46 (5%) | 7 (6%) |
| Male | 1487 (96%) | 719 (97%) | 833 (95%) | 113 (94%) |
| **Race** | | | | |
| Black | 253 (16%) | 110 (15%) | 142 (16%) | 20 (17%) |
| White | 1212 (78%) | 592 (80%) | 703 (80%) | 97 (81%) |
| Hispanic | 8 (1%) | 7 (1%) | 8 (1%) | 0 (0%) |
| Other | 15 (1%) | 8 (1%) | 3 (0%) | 1 (1%) |
| Not Reported | 64 (4%) | 25 (3%) | 23 (3%) | 2 (2%) |
| **BMI** | | | | |
| Median (Q1, Q3) | 27.7 (24.7, 31.7) | 27.5 (24.7, 30.7) | 28.3 (25.1, 32.6) | 28.8 (25.1, 32.3) |
| **BMI–categorized** | | | | |
| $<25$ kg/m$^2$ | 425 (27%) | 207 (28%) | 215 (25%) | 28 (24%) |
| 25–29.9 kg/m$^2$ | 585 (38%) | 316 (43%) | 328 (37%) | 39 (33%) |
| $\geq 30$ kg/m$^2$ | 540 (35%) | 219 (30%) | 334 (38%) | 51 (43%) |
| **Smoker** | | | | |
| Current | 321 (21%) | 125 (17%) | 169 (19%) | 18 (15%) |
| Ever | 910 (59%) | 468 (63%) | 399 (45%) | 76 (63%) |
| Never | 258 (17%) | 124 (17%) | 259 (29%) | 21 (18%) |
| Not Reported | 63 (4%) | 25 (3%) | 52 (6%) | 5 (4%) |
| **COPD** | 755 (49%) | 291 (39%) | 263 (30%) | 54 (45%) |
| **Asthma** | 117 (8%) | 49 (7%) | 82 (9%) | 6 (5%) |
| **Lung cancer** | 110 (7%) | 58 (8%) | 59 (7%) | 23 (19%) |
| **Gastroesophageal reflux disease** | 620 (40%) | 283 (38%) | 363 (41%) | 40 (33%) |
| **Coronary artery disease** | 515 (33%) | 298 (40%) | 250 (28%) | 43 (36%) |
| **Coronary heart failure** | 282 (18%) | 137 (18%) | 108 (12%) | 19 (16%) |
| **Stroke** | 172 (11%) | 84 (11%) | 68 (8%) | 11 (9%) |
| **Diabetes mellitus** | 575 (37%) | 282 (38%) | 306 (35%) | 35 (29%) |
| **Obesity** | 292 (19%) | 103 (14%) | 205 (23%) | 28 (23%) |
| **PFT group** | | | | |
| Not Reported | 829 | 391 | 879 | 56 |
| 1. Restrictive | 347 (48%) | 211 (60%) | 0 (0%) | 35 (55%) |
| 2. Obstructive | 245 (34%) | 71 (20%) | 0 (0%) | 15 (23%) |
| 3. Isolated DLCO Impairment | 108 (15%) | 63 (18%) | 0 (0%) | 12 (19%) |
| 4. Normal | 23 (3%) | 6 (2%) | 0 (0%) | 2 (3%) |

codes (excluding 516.31), and 83.6 months for 516.31. Notably, 515 and 516.31 had a lower survival rate than 516 subjects (excluding 516.31) (log-rank, p<0.001).

## Discussion

The present study identifies and describes a cohort of ILD subjects within the Veteran's Health Administration in the Mid-Atlantic Region. We embarked on this study to address if the veteran population is enriched for ILD. Our hypothesis was based on that fact that the veteran population exhibits risk factors (male sex, advanced age and active or prior smoking history) known to associate with increased ILD prevalence. In this study, we confirmed that ILDs are

**Table 2. Clinical care characteristics stratified by ICD-9 code.**

|  | 515 (N = 1552) | 516 (N = 742) | No ILD (N = 879) | Other ILD (N = 120) | p-value |
|---|---|---|---|---|---|
| **Had a high resolution CT scan** | 1347 (87%) | 589 (79%) | 637 (73%) | 110 (92%) | <0.001[1] |
| **Bronchoscopy** | 119 (8%) | 73 (10%) | 28 (3%) | 24 (20%) | <0.001[1] |
| **Biopsy type** |  |  |  |  | <0.001[1] |
| Not Reported | 6 | 2 | 1 | 0 |  |
| Surgery | 82 (5%) | 94 (13%) | 38 (4%) | 9 (8%) |  |
| TBBX | 76 (5%) | 47 (6%) | 23 (3%) | 12 (10%) |  |
| **Oxygen therapy at diagnosis** | 252 (16%) | 165 (22%) | 36 (4%) | 18 (15%) | <0.001[1] |
| **Oxygen therapy at last visit** | 629 (41%) | 436 (59%) | 173 (20%) | 47 (39%) | <0.001[1] |
| **PPI drug** | 942 (61%) | 454 (61%) | 510 (58%) | 79 (66%) | 0.285[1] |
| **Corticosteroid drug** | 758 (49%) | 334 (45%) | 268 (30%) | 51 (43%) | <0.001[1] |
| **Antifibrotic drug** | 18 (1%) | 46 (6%) | 0 (0%) | 0 (0%) | <0.001[2] |

[1]Chi-Square

[2]Fisher Exact

ICD- International Classification of Diseases; ILD–Interstitial Lung Disease; TBBX–Transbronchial Biopsy; PPI–Proton Pump Inhibitor

**Table 3.** A. Prevalence and incidence of ILD at six VA health care centers among patients with ICD-9 515 or 516. B. Prevalence and incidence of ILD at six VA health care centers among patients with ICD-9 515 or 516 and HRCT, surgical lung biopsy, or transbronchial lung biopsy.

**A.**

|  | Prevalence (per 100,000 people) | Incidence (per 100,000 person-years) | | | | | | | | |
|---|---|---|---|---|---|---|---|---|---|---|
|  | as of 12/31/15 | 2008 | 2009 | 2010 | 2011 | 2012 | 2013 | 2014 | 2015 | Average |
| **Durham** | 181 | 25 | 41 | 48 | 48 | 77 | 58 | 94 | 80 | 59 |
| **Fayetteville** | 106 | 18 | 32 | 39 | 38 | 46 | 38 | 45 | 29 | 36 |
| **Asheville** | 393 | 79 | 121 | 114 | 97 | 98 | 43 | 123 | 91 | 96 |
| **Richmond** | 240 | 40 | 101 | 73 | 81 | 81 | 56 | 135 | 70 | 80 |
| **Salem** | 395 | 59 | 96 | 64 | 77 | 88 | 65 | 98 | 80 | 78 |
| **Salisbury** | 138 | 20 | 147 | 99 | 111 | 117 | 78 | 158 | 108 | 105 |
| **Average** | 256 | 39 | 80 | 66 | 69 | 80 | 55 | 102 | 73 | 70 |

**B.**

|  | Prevalence (per 100,000 people) | Incidence (per 100,000 person-years) | | | | | | | | |
|---|---|---|---|---|---|---|---|---|---|---|
|  | as of 12/31/15 | 2008 | 2009 | 2010 | 2011 | 2012 | 2013 | 2014 | 2015 | Average |
| **Durham** | 167 | 19 | 40 | 39 | 45 | 76 | 56 | 83 | 70 | 53 |
| **Fayetteville** | 97 | 18 | 30 | 32 | 31 | 41 | 37 | 37 | 29 | 32 |
| **Asheville** | 379 | 85 | 97 | 96 | 91 | 92 | 51 | 123 | 89 | 91 |
| **Richmond** | 254 | 33 | 99 | 73 | 71 | 86 | 54 | 126 | 64 | 76 |
| **Salem** | 365 | 57 | 90 | 59 | 59 | 69 | 62 | 97 | 74 | 71 |
| **Salisbury** | 106 | 13 | 103 | 79 | 77 | 90 | 64 | 147 | 93 | 83 |
| **Average** | 237 | 35 | 69 | 57 | 57 | 72 | 52 | 95 | 67 | 63 |

Incidence per 100,000 subjects was calculated as the number of subjects diagnosed with an ILD in each year divided by the number of unique subjects who visited one of the included VA centers in that calendar year. Incidence rates were then averaged across years. Prevalence per 100,000 subjects was calculated as the number of subjects diagnosed with an ILD before December 31st, 2015 and still alive at this date, divided by the number of unique subjects who visited one of the included VA centers in 2013–2015 and were alive as of December 31st, 2015 (as an estimate of the number of VA subjects in the system at these centers).

ILD–Interstitial Lung Disease

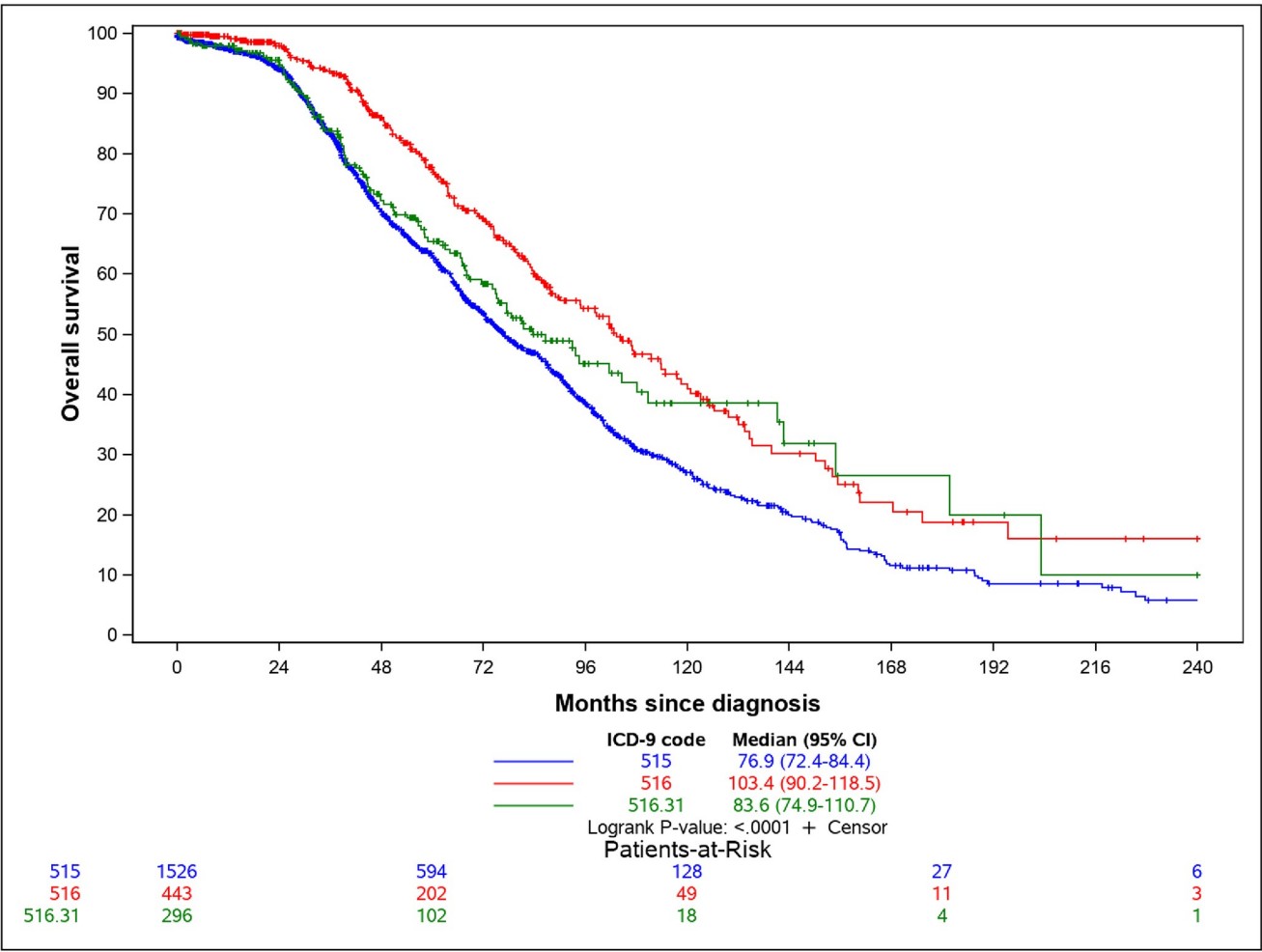

ICD- International Classification of Diseases; ILD – Interstitial Lung Disease; HR – Hazard Ratio; KM – Kaplan Meier

**Fig 1. Kaplan Meier survival based on ICD-9 code.** Information obtained from electronic medical record review was used to define survival for all-cause mortality.

enriched in our veteran cohort, including IPF. Additionally, we noted a number of individuals (26% of cohort) coded with an ILD diagnostic code, but in whom there was no observed ILD following chart review.

Our study identified an estimated prevalence of 256 per 100,000 people and an average incidence across the study period of 70 per 100,000 person-years for ILD subjects using ICD-9 codes and EMR manual review. We analyzed this data using broad and narrow case definitions as some veterans do not receive all of their care in the VA, and to address concerns about the accuracy of the diagnosis. Even with this narrow case definition, the ILD incidence and prevalence was significantly higher than other literature on ILD epidemiology [8]. Previous ILD registries in the USA have estimated prevalence rates of 14.3–63 per 100,000 people and incidence rates of 7.4–17.3 per 100,000 person-years [9–11]. European ILD epidemiological studies have also estimated rates much lower than this study with incidence rates ranging from 0.76–34.34 per 100,000 person-years [12–17].

Our findings confirm the prevailing hypothesis that ILD is enriched within the VA. Though not proven in the present study this is likely due to increased risk factors for ILD

in veterans. Whereas the prevalence of ever-smoking is less than 50% among the general population in the US, we found smoking rates to be more than 70%, consistent with enriched smoking rates in veterans [18]. Another potential explanation for the higher observed values is that our dataset evaluates a more generalized population than a specific registry. This is supported by recent studies using general population cohorts that demonstrated higher incident and prevalence rates than prior registry studies [19]. A limitation to our analysis is that the denominator we used to calculate incidence and prevalence was based on the number of patients who visited one of the included VA centers in a given time frame and did not include patients who are in the VA system but did not visit during that time period. Therefore, it is possible that we underestimated the total number of patients and therefore overestimated the incidence and prevalence rates. Our generous entry criteria (any individual with any ILD ICD-9 billing code during the defined study period) likely also identified more ILD subjects. Given that each of these underwent a manual EMR review, we are confident that this is not just a misclassification of individuals without ILD. Future studies will need to validate our observations in other VA regional networks and/or VA national databases.

We observed a significant population of subjects who had an ILD-associated billing code but no evidence of ILD following manual chart abstraction. This could suggest an inherent misunderstanding of the clinical criteria of ILD or coding based on an impression prior to obtaining imaging or other diagnostic studies. The extent of the miscoding was likely higher in our cohort due to our broad search criteria. This was purposely designed to be more inclusive in an attempt to capture as many individuals with ILD as possible. Despite this, our data are consistent with studies documenting inaccuracies in the use of billing codes for clinical research [20–22]. This issue was noted in recent a survey where more than half of the respondents with ILD had at least one clinical misdiagnosis [23]. Overall, our data highlights that caution should be used when defining ILD epidemiology solely based on cohorts identified by ICD codes without careful validation of their accuracy.

Interestingly, we observed a difference in survival between the 515 vs. 516 groups. Generally, IPF is considered the ILD with the highest mortality risk. As IPF falls under the 516 group (coded as 516.31), we expected that 516 mortality would be worse than those coded as 515. To address this, we also performed the analysis of the mortality of 516.31 separate from 516. The reason we observed worse mortality in the 515 group in our cohort is not clear. It is possible that IPF cases were misclassified as a 515 code. Similar to other epidemiological studies using ICD codes, we believe that the 515 code is sometimes used as the generic code for ILD, of which IPF is one of the most common diagnoses [11, 19, 24]. The Veteran population, based on demographics, has a pretest probability of IPF and thus even general 515 diagnostic codes may be capturing IPF. We observed a similar issue with use of the 516 codes, where the code for "IPF" (516.31) and the code for "other specified alveolar and parietoalveolar pneumonopathies" (516.8) exhibited very similar clinical characteristics, suggesting that the 516.8 code is likely a group that would be clinically diagnosed as IPF. Future studies will review specific clinical and radiographic criteria to evaluate miscoding between the 515 and 516 codes to understand the difference in mortality rates.

The comorbidities across the 515, 516, and other ILDs were similar except for higher rates of COPD in 515 codes. This is also reflected in the pulmonary function testing which showed a greater frequency of obstructed patterns in the 515 vs the other groups. Pulmonary function testing also noted a number of individuals with "isolated DLCO impairment" physiologic patterns. The high rates of obstructive and isolated DLCO impairment patterns could suggest a higher rate of co-existing emphysema, also known as combined pulmonary fibrosis and

emphysema [25]. This could be explained by the higher rates of smoking generally seen in veteran cohorts [18]. We also noted higher rates of coronary disease in 516 ICD 9 codes, which have been associated with progressive fibrotic diseases [26, 27].

We also defined the frequency of diagnostic testing and therapeutic interventions for ILD within our cohort to determine how often veterans received appropriate interventions. The American Thoracic Society, European Respiratory Society, and British Thoracic Society all provide best practice and guidelines on the diagnosis, treatment, and monitoring of subjects with ILD [28, 29]. High resolution computed tomography (HRCT) is critical for the initial diagnostic approach. Pulmonary function testing at diagnosis complements HRCT by providing a representation of severity. Furthermore, serial pulmonary function tests have prognostic value in ILD [30–33]. Despite relatively high performance of HRCT in this cohort, a significant proportion of the VA cohort did not receive PFTs, nor were these performed at later points as a measure to follow disease progression. A possible explanation is that many veterans receive healthcare principally through community-based providers and are only infrequently followed at the VA. Therefore, some of these studies may have been performed outside of the VA and were not captured in our database. Alternatively, it is possible that clinical care for ILD subjects in VA sites does not prioritize PFTs as an important clinical measure or that these patients have limited access to pulmonary providers with ILD expertise. Additionally, we noted low use of approved IPF therapies. A potential explanation is that the drugs were approved during the study period. Additionally, there was then a period of required central VA authorization of use and implementation at the individual VA sites in VISN6. Alternatively, this could reflect an unmet need in the VA, particularly at VISN6 sites which are not affiliated with academic medical centers where there is access to ILD specialists. In future work, we plan to explore differences by facility and available site resources.

Similar to other retrospective studies done on large datasets there are limitations to our analysis. First, our population only included United States military veterans from the Mid-Atlantic region, therefore this may not be generalizable to other VA sites. The veteran population is predominantly male, elderly, more-likely to be white, non-Hispanic, and economically dis-advantaged compared to the general population [34, 35]. Additionally, there may be a selection bias, as our cohort includes veterans who have opted to receive their medical care within the Veterans Health Administration. VHA users tend to be older, less economically advantaged, report more chronic medical conditions, and have higher rates of combat exposure than non-VHA users [36, 37]. We did not acquire data from outside the VHA health system for veterans that receive care through the community (i.e. non-VA community care). Prior studies of the veteran's population has shown that two-thirds of veterans have access to non-VHA care through other government programs and that one-half of these veterans will receive care outside the VHA [38]. Veterans can also be sent to community providers through various VHA programs if there is a lack of specific ILD specialty providers or long wait times. Third, our study lacked age-matched controls to use a comparator group. Fourth, we relied on billing codes from a single encounter [39] to find our initial cohort, which, as noted above, can be inaccurate. The decision to use a single encounter was made to collect a population as comprehensive as possible with the limitation that this would likely increase our coding inaccuracies. Lastly, veteran deaths outside the VHA system are not generally recorded in the VHA data warehouse, thus potentially affecting our ability to accurately define mortality. Despite these limitations, we believe our results provide insight into the real-world epidemiology of interstitial lung disease and its impact on veterans' health.

## Conclusion

Here we report the epidemiology of a large distinct cohort of VHA patients with interstitial lung disease. Identifying patients by diagnostic codes followed by a detailed EMR review allow us, for the first time, to define the types and characteristics of ILD in a VHA population. Based on this analysis, veterans are at a substantially higher risk for developing ILDs than other non-VA cohorts. It highlights that ILD is a critical issue for veterans' health, and requires increased attention and awareness for ILD within the VA.

## Supporting information

**S1 Data.**
(CSV)

**S1 Table. ICD-9 codes.**
(JPG)

**S2 Table. Characteristics of patient population.**
(JPG)

**S3 Table. Characteristics of patient population stratified by ICD-9 code.**
(JPG)

**S1 Fig. CONSORT diagram of the VISN6 ILD cohort based on ICD-9 code.**
(JPG)

## Acknowledgments

JCB performed the initial subject identification and data extraction. RK assisted with data warehouse development, data cleaning, and data extraction. AB, RAP, RMT and AR performed chart abstraction and review for quality. LH performed the statistical analysis. DS and KW assisted with data interpretation. AB and RMT drafted the manuscript. RAP and RMT conceived of the overall project.

## Author Contributions

**Conceptualization:** Roy A. Pleasants, Karen Welty-Wolf, Robert M. Tighe.

**Data curation:** Armando Bedoya, Roy A. Pleasants, Joel C. Boggan, Danielle Seaman, Anne Reihman, Robert Kundich, Robert M. Tighe.

**Formal analysis:** Armando Bedoya, Roy A. Pleasants, Joel C. Boggan, Danielle Seaman, Anne Reihman, Lauren Howard, Robert Kundich, Robert M. Tighe.

**Funding acquisition:** Roy A. Pleasants, Robert M. Tighe.

**Investigation:** Lauren Howard, Robert M. Tighe.

**Methodology:** Lauren Howard.

**Project administration:** Robert M. Tighe.

**Resources:** Joel C. Boggan, Robert Kundich.

**Software:** Robert Kundich.

**Supervision:** Karen Welty-Wolf, Robert M. Tighe.

**Validation:** Armando Bedoya, Lauren Howard, Robert Kundich, Robert M. Tighe.

**Visualization:** Armando Bedoya, Danielle Seaman, Lauren Howard, Robert Kundich.

**Writing – original draft:** Armando Bedoya, Robert M. Tighe.

**Writing – review & editing:** Armando Bedoya, Roy A. Pleasants, Joel C. Boggan, Danielle Seaman, Anne Reihman, Lauren Howard, Robert Kundich, Karen Welty-Wolf, Robert M. Tighe.

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
