## [Decision Letter · Decision Letter 0]

7 Dec 2020

PONE-D-20-27587

Interstitial Lung Disease in a Veterans Affairs Regional Network; a Retrospective Cohort Study

PLOS ONE

Dear Dr. Tighe,

Thank you for submitting your manuscript to PLOS ONE. After careful consideration, we feel that it has merit but does not fully meet PLOS ONE’s publication criteria as it currently stands. Therefore, we invite you to submit a revised version of the manuscript that addresses the points raised during the review process.

We look forward to receiving your revised manuscript.

Kind regards,

Mehrdad Arjomandi, MD

Academic Editor

PLOS ONE

Journal Requirements:

2. In your ethics statement in the manuscript and in the online submission form, please provide additional information about the patient records used in your retrospective study. Specifically, please ensure that you have discussed whether all data were fully anonymized before you accessed them.

3. Please include your full ethics statement in the Methods.

4.Thank you for stating the following in the Competing Interests section:

[Dr. Tighe discloses that he has received funding from the NIH and industry to perform research studies

unrelated to the present study. He also has served on an advisory board for Boehringer Ingelheim. None of

the other authors have any conflicts of interest to report for the submitted work. ].

Reviewers' comments:

Reviewer's Responses to Questions

**Comments to the Author**

1. Is the manuscript technically sound, and do the data support the conclusions?

Reviewer #1: Partly

Reviewer #2: Yes

2. Has the statistical analysis been performed appropriately and rigorously? 

Reviewer #1: N/A

Reviewer #2: Yes

3. Have the authors made all data underlying the findings in their manuscript fully available?

Reviewer #1: Yes

Reviewer #2: Yes

4. Is the manuscript presented in an intelligible fashion and written in standard English?

Reviewer #1: Yes

Reviewer #2: Yes

5. Review Comments to the Author

Reviewer #1: General: This is an interesting study that seeks to establish the burden of ILD in the Veteran population. The comments made below are for additional clarification. I hope that the authors find them useful.

Abstract

1. Incidence should be reported as person-time (example per 100,000 person-years)

2. It would be helpful to also include incidence and prevalence as a percentage in parenthesis. For example, 256 per 100,000 (0.26%).

Introduction

1 The intro refers to morbidity mortality of 3.75 per 100,000 people. It is unclear what this is referencing – IPF, ILD? As an attributable fraction mortality of the general population? Please clarify. It seems like a low morbidity and mortality rate for IPF.

2 The authors report that the incidence and prevalence of ILD is poorly characterized. This is not quite accurate as there has been substantial epidemiologic work in this area. However, as the authors point out, not much has been published from the VA. I would suggest reframing paragraph 2 of the introduction in a different way – that there is a gap in knowledge about epidemiology of ILD in the VA. As it is currently written, paragraph 2, line 84 suggests that the VA can fil the general ILD epidemiology knowledge gap. This is not necessarily true as the VA is a very unique population that may not be generalizable to other populations.

Methods

1. The ICD-9 to ICD-10 conversion happened in 10/2015. The authors refer to study period as 1/1/2008 to 12/31/2015 – please clarify

2. Additional discussion on how the ILD ICD codes were selected would be helpful. Prior literature references? Understanding the authors algorithm approach would be helpful to the reader.

3. Please include details about how comorbidity data, bronchoscopy, CT scan etc was extracted … was it by ICD/CPT code? If so, include those codes in the supplement. What was the look back period for comorbidities (ex. 2 outpatient comorbidity codes over the year prior to ILD diagnosis).

4. For period prevalence from 2013 – 2015, it seems like the authors excluded patients who died before 12/31/2015. If I am understanding this correctly, that means that someone who was diagnosed with ILD and died 11/2015 would not be counted in the period prevalence calculation? Thus, their contribution to prevalence for 2013 and 2014 would be lost? Is that correct? This seems like it would lead to under estimation of prevalence. Please clarify.

5. For incidence calculation, did authors exclude patients who had prior ILD ICD codes in previous years? What was the lookback period?

6. More discussion is needed on chart review. How was the diagnosis of ILD confirmed on chart review? Physician note? Multidisciplinary conference? CT with fibrosis on report?

7. Could patients who had 515 codes then go on to have 516.3 codes in subsequent years? In other words, could the same patient be part of multiple different samples? This would be important to note if the cohorts outlined are or are not mutually exclusive.

8. The authors report that in many cases, there was no PFT data recorded in EMR. Did authors look at PIT (non-VA community care paid for by the VA) data?

Results

1. For Figure E1 (CONSORT diagram), it would be helpful to know what codes were represented in the “no ILD on chart review” and #/% breakdown.

2. The low use of antifibrotic does not seem appropriate for this study – pirfenidone and nintedanib were approved towards the end of the study period and thus the low rates of utilization may be biased as only a very small fraction would have been eligible.

3. Please see questions about incidence and prevalence calculations above in methods section.

4. It appears that the authors did quite extensive chart review, which is quite impressive. However, something that is missing from the results is a discussion of the accuracy of the diagnosis when charts were reviewed. Although the authors do account for yes/no ILD, there seems to be an opportunity to also state whether the ILD codes used reflect specific ILD diagnostic accuracy. For example, did someone diagnosed with an ICD code for sarcoid truly have sarcoid? Is this data available?

5. For patients who had no ILD on chart review, could it be because they received part of their care outside the VA?

6. For those who did not have ILD but had an ICD code for ILD, it would be helpful to provide additional details here about how that misclassification occurred. What were those ICD codes associated with? Imaging? Miscoding on outpatient notes?

Discussion

1. Please specify in the first sentence that the study represents the Mid-Atlantic region.

2. The authors state in line 245 that they had a large number of individuals coded with ILD but with no observed ILD on chart review. I would suggest removing the world large and specify the percentage. 26% misclassification is actually lower than what has been noted in some other databases which have reported up to 50% misclassifications.

3. Lines 285 – 287 seem somewhat misplaced as it does not seem like the authors were specifically evaluating diagnostic disagreement at the clinical level.

4. I would suggest reframing the discussion lines 290 – 302 about 515 vs 515.3. Other studies have suggested that the 515 code is often given in the initial workup of IPF and many patients with IPF may have only these diagnosis codes. My suggestion would be to frame it as in the Veteran population, based on demographics, there is a high pre test probability of IPF and thus even general 515 diagnostic codes may be capturing IPF.

5. Could the possibility that CPFE may be captured by the 515 code explain the difference in survival between the cohorts? Does this represent a unique subgroup of patients with IPF + emphysema?

6. The comment about Veteran death not being recorded in VHA data warehouse is surprising. As I understand it, death date is captured in CDW even if the patient’s death occurred outside the VA system. Please confirm.

7. One additional limitation to mention is that the data includes one geographic region and may not be generalizable to the rest of the US (ex. due to differences in exposures, age breakdown etc.)

Reviewer #2: The manuscript described the incidence/prevalence, clinical characteristics and outcomes of ILD patients within veteran’s administration Mid Atlantic Health Care Network (VISN6) from January 1, 2008 to December 31st, 2015. Patients were identified by at least one visit encounter with a 515, 516, or other ILD ICD-9 cod. The study identified 3293 subjects met the inclusion criteria. 879 subjects (26%) had no evidence of ILD following manual medical record review. Overall estimated prevalence in verified ILD subjects was 256 per 100,000 people with a mean incidence across the years of 70 per 100,000 people. The prevalence and mean incidence when focusing on people with an ILD diagnostic code who had a HRCT scan or a bronchoscopic or surgical lung biopsy was 237 per 100,000 people and 63 per 100,000 people respectively. The median survival was 76.9 months for 515 codes, 103.4 months for 516 codes, and 83.6 months for 516.31. The study concluded from this retrospective cohort data that there are high ILD incidence/prevalence within the VA. Therefore, ILD is an important VA health concern.

General comments

I believe that this is an interesting study with comprehensive data of ILD among Veteran population. There are some difficulties in conducting such a study which I believe the authors have addressed reasonably well. Prospective study addressing ILDs among Veteran population is highly needed to address multiple question which was difficult to address in the current study.

Major Critiques:

1) Certain types of ILD like chronic hypersensitivity pneumonitis and connective disease (CTD-ILD) which represents a large portion of ILD among Veteran was not addressed in current study

2) Inhalational and environmental exposure during military deployment is very common among Veteran and consider a risk for developing ILDs. The authors did not address veteran occupational and environmental exposure.

3) It is not clear if autoimmune work up for CTD was part of ILD work up in this ILD Cohort

4) HRCT chest was done in large number of ILD patients with Code 515 (87%), 516 (73%), other ILD (92%). It would be interesting if the author would use chest HRCT data to define the type of ILD and correlate this data with ICD-9 code.

Minor Critiques:

Abstract/introduction

1) There is contradiction between research question in the abstract and primary objective of this study described in last 3 lines of the introduction section “ The primary objective of this study is to describe the characteristics, diagnosis, management, and outcomes of the patients with ILD”

The management of ILD patient was not clearly described in the current study as most of these ILD patients was managed outside VA systems. The study did not discuss for example steroid/ Azathioprine and N-acetyl-cysteine which were commonly used in the period from 2008-2014. The study did not discuss the use of pulmonary rehabilitation and lung transplant as modalities of ILD treatment.

Result section

1) Line 185-186: There was a moderate use of corticosteroids and low rates of IPF therapy use across the cohort. Please explain low rate of IPF therapy use (antifibrotics or other).

2) The study examined ILD patients from 2008 till 2015 where most of patient were treated with corticosteroid and Azathioprine and N-acetylcysteine. Since antifibrotics drugs (Pirfenidone and Nintedanib) were approved by FDA in October 2014 for use in IPF. It is possible that low survival rate due to the use of steroid and lack of antifibrotic therapy.

3) Table E2: Mixed connective tissue disease was present in 1% of entire cohort. Please explain the association between CTD and ILD.

Discussion

Line 294-295: “it is possible that IPF cases were misclassified as a 515 code” To confirm this statement, is any possibility the authors look to the clinical and radiologic data to confirm the diagnosis of ICD-9 code especially the authors were able to exclude 879 subjects (26%) of cohort after finding no radiologic and clinical evidence of ILD.

6. PLOS authors have the option to publish the peer review history of their article (what does this mean?). If published, this will include your full peer review and any attached files.

Reviewer #1: No

Reviewer #2: **Yes: **Nadia A Hasaneen

---

## [Author Response · Author response to Decision Letter 0]

7 Jan 2021

Response to Reviewer Comments to the Author

Reviewer #1: 

General: This is an interesting study that seeks to establish the burden of ILD in the Veteran population. The comments made below are for additional clarification. I hope that the authors find them useful.

a. We thank the review for the positive appreciation of the study. 

Abstract

1. Incidence should be reported as person-time (example per 100,000 person-years) 

a. We have made this change in the manuscript to address this concern. All incidence was calculated per year (with one year being the interval). Therefore, the data per 100,000 people and per 100,000 person-years is equivalent in this setting. 

2. It would be helpful to also include incidence and prevalence as a percentage in parenthesis. For example, 256 per 100,000 (0.26%). 

a. We added the percentages in parenthesis to the text.

Introduction

1. The intro refers to morbidity mortality of 3.75 per 100,000 people. It is unclear what this is referencing – IPF, ILD? As an attributable fraction mortality of the general population? Please clarify. It seems like a low morbidity and mortality rate for IPF.

a. We apologize for the confusion. This was a reference to males with IPF. It came from a 2018 study looking at WHO data of European Union IPF data from 2001-2013. We clarified this in the text specifying that this refers to median mortality of IPF in males standardized to the 2013 European population. 

2. The authors report that the incidence and prevalence of ILD is poorly characterized. This is not quite accurate as there has been substantial epidemiologic work in this area. However, as the authors point out, not much has been published from the VA. I would suggest reframing paragraph 2 of the introduction in a different way – that there is a gap in knowledge about epidemiology of ILD in the VA. As it is currently written, paragraph 2, line 84 suggests that the VA can fil the general ILD epidemiology knowledge gap. This is not necessarily true as the VA is a very unique population that may not be generalizable to other populations. 

a. We appreciate the feedback and have reformatted the text to try and address the concern. We do feel that there is a relative paucity of epidemiology data on ILD. The majority of this data focuses exclusively on IPF or on specific types of ILD (though to a lesser extent in these subgroups). Additionally, the data from ILD tends to be from single site centers. Therefore, larger databases such as the VA may offer some additional information on ILD epidemiology that is not presently available. We do agree with the reviewer that the veteran population is unique and therefore has some unique characteristics from a generalized population. We highlighted this in the limitations of the manuscript. However, we also feel that the data available from the VA including nationwide VA data could offer insights about ILD overall that are presently not available from single center and/or Medicare claims databases. 

Methods

1. The ICD-9 to ICD-10 conversion happened in 10/2015. The authors refer to study period as 1/1/2008 to 12/31/2015 – please clarify. 

a. We studied subjects who were coded with ILD with ICD-9 during the aforementioned years, then followed them forward through till January 2019. We did not capture ICD10 codes from 10/01/2015-12/31/2015. This is now clarified in the manuscript.

2. Additional discussion on how the ILD ICD codes were selected would be helpful. Prior literature references? Understanding the authors algorithm approach would be helpful to the reader. 

a. The goal of the present study was to be inclusive of any ILD ICD-9 codes. Our experimental design was to identify any potential patients with ILD and then use the chart review to clarify the diagnosis. The ILD ICD codes were selected by the research team, primarily pulmonologists who specialize in ILD, based upon review of the available ICD-9 codes for ILD. We did not restrict ILD ICD-9 codes for this study. This is now clarified in the text.

3. Please include details about how comorbidity data, bronchoscopy, CT scan etc was extracted … was it by ICD/CPT code? If so, include those codes in the supplement. What was the look back period for comorbidities (ex. 2 outpatient comorbidity codes over the year prior to ILD diagnosis)?

a. Comorbidities, procedures, and imaging were principally extracted from the corporate data warehouse through ICD/CPT codes. These were then confirmed during the manual chart abstraction, and comorbidities were added if noted in the progress notes but not specifically coded as an ICD (e.g. GERD). To be as inclusive as possible, there was no limit to the lookback period as long as there were available records in the VA EMR. The prospective data collection went through January 2019 when the database was locked. 

4. For period prevalence from 2013 – 2015, it seems like the authors excluded patients who died before 12/31/2015. If I am understanding this correctly, that means that someone who was diagnosed with ILD and died 11/2015 would not be counted in the period prevalence calculation? Thus, their contribution to prevalence for 2013 and 2014 would be lost? Is that correct? This seems like it would lead to under estimation of prevalence. Please clarify. 

a. We appreciate the close review and apologize for the confusion, we did not restrict mortality events. We recorded all deaths that occurred between 2013 and January 2019 for all subjects if the data was available. This is now clarified in the methods section.

5. For incidence calculation, did authors exclude patients who had prior ILD ICD codes in previous years? What was the lookback period? 

a. We did not exclude patients who had prior ILD codes in the previous years. Once identified a subject with at least one ILD encounter, we went back as far as possible in CPRS for those subjects to collect data. Therefore, in some incidences the subject was identified in the study period by and encounter but their date of diagnosis was before the specified period. The purpose of defining individuals this way not to limit the cohort but rather to define all individuals with an ILD encounter code and then define the characteristics of those individuals regardless of the date of their initial diagnosis, etc. We hope this clarifies the reviewer’s concern and discussion about this was added to the methods section.

6. More discussion is needed on chart review. How was the diagnosis of ILD confirmed on chart review? Physician note? Multidisciplinary conference? CT with fibrosis on report? 

a. We agree that this is an important consideration particularly since we hypothesized that there could be significant inaccuracy between the diagnostic code and the actual ILD diagnosis. Therefore, the ICD-9 diagnosis was confirmed based on a thorough chart review. The data from this chart review included physician notes, and CT scan reports. This was reviewed by the study team to confirm or refute the diagnosis. As the study included VA’s not associated with academic centers, there was not universal availability of a multidisciplinary conference for review of the ILD diagnosis at these sites. We have clarified this in the text. 

7. Could patients who had 515 codes then go on to have 516.3 codes in subsequent years? In other words, could the same patient be part of multiple different samples? This would be important to note if the cohorts outlined are or are not mutually exclusive. 

a. There were instances where patients were coded with both 515 and 516 codes. In this setting, the study team used the physician notes, exam, laboratory values and CT scan report to clarify which of the ICD-9 codes were accurate and this was then used this to define their ICD-9 group. This typically occurred in the 515 group and led to either refinement to a 516 group or determination that the subject did not have an ILD (typically miscoded based on being lung granuloma). No individual ended up in two cohorts in this study. To address this potential confusion, we have clarified this in the text.

8. The authors report that in many cases, there was no PFT data recorded in EMR. Did authors look at PIT (non-VA community care paid for by the VA) data? 

a. We thank the reviewer for the astute observation. We did not look at PIT data. We acknowledge this as a limitation of our study in the Discussion section. Unfortunately, we did not have complete access to the documentation (notes, results, imaging) of non-VA community care. As these records were not consistently available and we were concerned about including potentially limited and inaccurate datasets.

Results

1. For Figure E1 (CONSORT diagram), it would be helpful to know what codes were represented in the “no ILD on chart review” and #/% breakdown.

a. We thank the reviewer for this suggestion. We have now updated this graphic to include this information.

2. The low use of antifibrotic does not seem appropriate for this study – pirfenidone and nintedanib were approved towards the end of the study period and thus the low rates of utilization may be biased as only a very small fraction would have been eligible. 

a. We appreciate the astute observation. Though pirfenidone and nintedanib were approved towards the end of the study period, there was an additional delay as the VA went through a process of authorization of use and then the local sites had to develop the appropriate processes for distribution. We agree that this could introduce bias. We now add this concern to the Discussion. However, we feel it is important to keep this information in the manuscript as it will inform if this will increase over time as they are available in the VA and also if there are differences in use based on regional VA sites despite a known population of veterans with ILDs qualifying their use. 

3. Please see questions about incidence and prevalence calculations above in methods section.

a. This has been addressed above

4. It appears that the authors did quite extensive chart review, which is quite impressive. However, something that is missing from the results is a discussion of the accuracy of the diagnosis when charts were reviewed. Although the authors do account for yes/no ILD, there seems to be an opportunity to also state whether the ILD codes used reflect specific ILD diagnostic accuracy. For example, did someone diagnosed with an ICD code for sarcoid truly have sarcoid? Is this data available? 

a. The reviewer raises an excellent point. The present study focused on the overall characteristics of the cohort and asked the initial question of how often ILD was coded but did not have clinical evidence supporting an ILD diagnosis. For this, we relied on the physician notes and CT scan reports to assess the accuracy of a yes/no for the diagnosis. The question of the diagnostic accuracy of specific ILD subtype is one we are very interested in assessing. This specific question is a focus of future analysis that is being presently performed. This will focus on the accuracy of the original physician diagnosis with centralized reviews of CT scans, etc. The data is available, but is beyond the scope of this initial report of our dataset.

5. For patients who had no ILD on chart review, could it be because they received part of their care outside the VA? 

a. We agree that this is a possibility and acknowledged it as a potential limitation. A large number of these individuals had some clinical data typically including radiographs that did not support an ILD diagnosis. However, it is possible that the reviewer is correct. 

6. For those who did not have ILD but had an ICD code for ILD, it would be helpful to provide additional details here about how that misclassification occurred. What were those ICD codes associated with? Imaging? Miscoding on outpatient notes?

a. We agree with the reviewer. Unfortunately, we have limited data for these patients. The original extraction used encounter codes to identify these patients. However, in general, the patients lacked radiographic data or outpatient notes supporting the use of the encounter codes. Alternatively, in several cases the ILD encounter code was associated with a radiograph revealing lung granulomas on the report. We expected miscoding based on our study design, as the goal was to identify as many ILD as possible and then confirm accurate diagnosis based on the chart review.

Discussion

1. Please specify in the first sentence that the study represents the Mid-Atlantic region.

a. We have added this clarification.

2. The authors state in line 245 that they had a large number of individuals coded with ILD but with no observed ILD on chart review. I would suggest removing the world large and specify the percentage. 26% misclassification is actually lower than what has been noted in some other databases which have reported up to 50% misclassifications.

a. We thank the reviewer for this comment. We have made this change.

3. Lines 285 – 287 seem somewhat misplaced as it does not seem like the authors were specifically evaluating diagnostic disagreement at the clinical level.

a. We appreciate the feedback and have made the change.

4. I would suggest reframing the discussion lines 290 – 302 about 515 vs 515.3. Other studies have suggested that the 515 code is often given in the initial workup of IPF and many patients with IPF may have only these diagnosis codes. My suggestion would be to frame it as in the Veteran population, based on demographics, there is a high pretest probability of IPF and thus even general 515 diagnostic codes may be capturing IPF.

a. We agree with the review. Our pretest probability was that there would be IPF patients coded as 515. During the course of the chart review, if it was determined that they did have IPF, then this was converted to the appropriate code. However, there could still be inaccuracy based on the clinical and radiologic interpretation of the treating physicians. We appreciate the feedback and have made that change to clarify this concern. 

5. Could the possibility that CPFE may be captured by the 515 code explain the difference in survival between the cohorts? Does this represent a unique subgroup of patients with IPF + emphysema? 

a. We are uncertain about the reason for the difference in the survival between the cohorts. It was not an expected finding. We did observe CPFE in the VA ILD cohort. The reviewer asks an interesting question and one we have actively been working on. We are currently performing an analysis of CPFE and hopefully will publish on this question soon. We felt that it was beyond the scope of this initial manuscript.

6. The comment about Veteran death not being recorded in VHA data warehouse is surprising. As I understand it, death date is captured in CDW even if the patient’s death occurred outside the VA system. Please confirm. 

a. Death is a difficult topic in the Electronic Health Record. The federal government attempts to track all deaths across the US via its own mechanisms (social security) but also through state mechanisms. The VHA receives various files but not all death files are uploaded into the CDW. There can also be a delay in when this information becomes available in the HER. 

7. One additional limitation to mention is that the data includes one geographic region and may not be generalizable to the rest of the US (ex. due to differences in exposures, age breakdown etc.) 

a. We agree that this is a potential concern. Future studies looking at larger datasets will help to address this question. We have added this as a limitation to the text. 

Reviewer #2: 

The manuscript described the incidence/prevalence, clinical characteristics and outcomes of ILD patients within veteran’s administration Mid Atlantic Health Care Network (VISN6) from January 1, 2008 to December 31st, 2015. Patients were identified by at least one visit encounter with a 515, 516, or other ILD ICD-9 cod. The study identified 3293 subjects met the inclusion criteria. 879 subjects (26%) had no evidence of ILD following manual medical record review. Overall estimated prevalence in verified ILD subjects was 256 per 100,000 people with a mean incidence across the years of 70 per 100,000 people. The prevalence and mean incidence when focusing on people with an ILD diagnostic code who had a HRCT scan or a bronchoscopic or surgical lung biopsy was 237 per 100,000 people and 63 per 100,000 people respectively. The median survival was 76.9 months for 515 codes, 103.4 months for 516 codes, and 83.6 months for 516.31. The study concluded from this retrospective cohort data that there are high ILD incidence/prevalence within the VA. Therefore, ILD is an important VA health concern.

General comments

I believe that this is an interesting study with comprehensive data of ILD among Veteran population. There are some difficulties in conducting such a study which I believe the authors have addressed reasonably well. Prospective study addressing ILDs among Veteran population is highly needed to address multiple question which was difficult to address in the current study.

a. We completely agree with the reviewer and appreciate their favorable comments. We always viewed the present study as being the first of many that are needed to understand ILD within the VA. 

Major Critiques:

1. Certain types of ILD like chronic hypersensitivity pneumonitis and connective disease (CTD-ILD) which represents a large portion of ILD among Veteran was not addressed in current study.

a. We agree with the reviewer. We are presently formulating a manuscript that focuses specifically on CTD-ILD in our cohort to expand on this present study. Hypersensitivity Pneumonitis diagnosis (ICD9 495.9) is included in the “other category” in the present study. We agree that chronic hypersensitivity pneumonitis is common in veterans (based on our own clinical experience) and this is not well documented in the present dataset. However, this diagnosis is complicated and frequently overlaps with IPF. Accurate distinction requires a level of detail and expertise that is not present in the available clinic notes and radiology studies (for example, many of CT scans are high resolution but not with inspiratory and expiratory imaging). Future studies will need to consider understanding this in the VA but will likely need to be prospective studies with specific pulmonary and radiologic expertise.

2. Inhalational and environmental exposure during military deployment is very common among Veteran and consider a risk for developing ILDs. The authors did not address veteran occupational and environmental exposure.

a. We agree this is an important topic, particularly in Veteran populations. We do have this data presently but are planning to address this issue in a subsequent manuscript.

3. It is not clear if autoimmune work up for CTD was part of ILD work up in this ILD Cohort

a. We agree this is an important point about accurate diagnosis of a CTD-ILD. One of the overriding issues we feel this manuscript and future ones will demonstrate is the lack of consistency in the diagnostic evaluation of ILD within the VA. As a result, autoimmune work-ups are inconsistently performed depending on the regional site and the individual providers. This makes it difficult to provide a definitive answer to the reviewer’s important question. 

4. HRCT chest was done in large number of ILD patients with Code 515 (87%), 516 (73%), other ILD (92%). It would be interesting if the author would use chest HRCT data to define the type of ILD and correlate this data with ICD-9 code. 

a. We agree completely with the reviewer. This first manuscript was limited to the specific questions outlined. However, we agree this is an important question. A portion of the CT scans in this cohort were independently reviewed by a chest radiologist to evaluate their accuracy. This is a planned analysis that is currently in process.

Minor Critiques:

Abstract/introduction

1. There is contradiction between research question in the abstract and primary objective of this study described in last 3 lines of the introduction section “The primary objective of this study is to describe the characteristics, diagnosis, management, and outcomes of the patients with ILD”

a. We thank the reviewer for picking up this discrepancy. We have corrected this in the text.

2. The management of ILD patient was not clearly described in the current study as most of these ILD patients was managed outside VA systems. The study did not discuss for example steroid/ Azathioprine and N-acetyl-cysteine which were commonly used in the period from 2008-2014. The study did not discuss the use of pulmonary rehabilitation and lung transplant as modalities of ILD treatment. 

a. The reviewer asks an important question. Much of the ILD care occurred within the VA in this cohort. As noted in Figure 2, there was steroid use in this cohort (though the majority did not appear to be sustained use but rather short steroid courses). In terms of azathioprine use, this was low in the VA cohort (~3%) with similar low percentage of use of mycophenolate. This suggests low use of steroid sparing agents and other measures common to ILD management in the VA. This includes lung transplant, which was very uncommon in the cohort. One of the issues we hope that our study and future studies will raise is the importance of ILD within the VA, thereby guiding efforts to improve the care of this disease. 

Result section

1. Line 185-186: There was a moderate use of corticosteroids and low rates of IPF therapy use across the cohort. Please explain low rate of IPF therapy use (antifibrotics or other).

a. Much of the steroid use was for shorter courses of therapy (acute steroid tapers). However, it was difficult to distinguish these during the course of the chart review unless specifically spelled out by the clinician in the chart. We agree that the use of IPF therapies was low in this cohort. This was discussed in response to reviewer #1. We suspect this is a combination of the timing of the cohort (ala around the time of initial IPF therapy approval), delayed access to the medication and poor identification and treatment of IPF. We feel this data is important to raise this as a potential area of future research within the VA. We added a discussion about the low rates of IPF therapies in the Discusion section.

2. The study examined ILD patients from 2008 till 2015 where most of patient were treated with corticosteroid and Azathioprine and N-acetylcysteine. Since antifibrotics drugs (Pirfenidone and Nintedanib) were approved by FDA in October 2014 for use in IPF. It is possible that low survival rate due to the use of steroid and lack of antifibrotic therapy.

a. This is an excellent question. We agree that this cohort was developed during a period of transition in IPF care. As the results of the PANTHER trial supported that immunosuppressive regimens were associated with harm, while pirfenidone and nintedanib were approved as a treatment for IPF. This raises questions about the difference in survival rates. It is possible that this could account for the differences observed. We suspect that it reflects issues of treatment, diagnostic accuracy and appropriate care. Despite our best efforts, this was not able to be clarified in the cohort. We feel this data will highlight questions about the use of IPF and ILD treatments in the VA and direct future studies. 

3. Table E2: Mixed connective tissue disease was present in 1% of entire cohort. Please explain the association between CTD and ILD. 

a. We apologize for the confusion. For the sake of clarity, we removed this mixed connective tissue designation from the table. We are presently drafting a separate manuscript focused on individuals diagnosed with CTD-ILD and their characteristics.

Discussion

1. Line 294-295: “it is possible that IPF cases were misclassified as a 515 code” To confirm this statement, is any possibility the authors look to the clinical and radiologic data to confirm the diagnosis of ICD-9 code especially the authors were able to exclude 879 subjects (26%) of cohort after finding no radiologic and clinical evidence of ILD. 

a. In the present study, we used the available clinical records, PFTs and radiology reports to assess the rationale behind the diagnostic code utilized. If there was a diagnosis of IPF in this evaluation, but a miscoding as 515 in this setting the record was corrected. However, we did not independently review the CT scan for this present study to assess this accuracy. For this reason, we agree that it is possible that there may be additional cases of IPF in the 515 group. Alternatively, for the no ILD group this assessment was possible based on the available data in the electronic record. We hope this clarifies the reviewer’s concern.

---

## [Editor Report · Decision Letter 1]

5 Feb 2021

Interstitial Lung Disease in a Veterans Affairs Regional Network; a retrospective cohort study

PONE-D-20-27587R1

Dear Dr. Tighe,

We’re pleased to inform you that your manuscript has been judged scientifically suitable for publication and will be formally accepted for publication once it meets all outstanding technical requirements.

Kind regards,

Mehrdad Arjomandi, MD

Academic Editor

PLOS ONE
---

## [Editor Report · Acceptance letter]

4 Mar 2021

PONE-D-20-27587R1 

Interstitial Lung Disease in a Veterans Affairs Regional Network; a retrospective cohort study 

Dear Dr. Tighe:

I'm pleased to inform you that your manuscript has been deemed suitable for publication in PLOS ONE. Congratulations! Your manuscript is now with our production department. 

Kind regards, 

on behalf of

Dr. Mehrdad Arjomandi 

Academic Editor

PLOS ONE